# Rotation-Equivariance and Position Encodings for Enhancing Local Descriptors

## Abstract

Keypoint extraction and description are crucial issues in robot vision. In recent years, deep learning based keypoint extraction have exhibited robustness to variations in lighting and viewpoint. However, due to the lack of rotational invariance in traditional convolutional networks, performance of deep learning-based keypoint significantly deteriorates under large rotations. Group-equivariant neural networks based Keypoint address the issue of rotational equivariance, but their overall performance also suffers. This paper addresses the problem from the perspective of keypoint description and proposes a fusion of locally rotation-equivariant descriptions with globally encoded positional information and a directional uncertainty weighted descriptor loss. This effectively enhances the performance of keypoint extraction and description. Validation is conducted on rotated-HPatches, rotated-MegaDepth and rotated-YFCC100M datasets.

## 1 Introduction

A keypoint extractor is a critical component in the field of computer vision, designed to identify and describe significant local image features within an image. Ma et al. (2021). It comprises two primary components: keypoint detection and keypoint description. Keypoint detection is responsible for locating prominent local regions in the image, typically characterized by unique textures, corners, or edges. These detected keypoints possess the following characteristics: they are conspicuous within the image, exhibit stability across different scales and lighting conditions, and can be accurately matched across different images Lowe (1999; 2004); Tang et al. (2019). The task of keypoint description is to encode the local surroundings of each detected keypoint into numerical descriptors. These keypoint extractor can be employed in various computer vision tasks such as image matching, 3D reconstruction, and object tracking Jiang et al. (2021).

In most application scenarios, rotational invariance is indeed not as important, such as visual odometers for low speed autonomous driving, visual position recognition, and object tracking. However, when these problems are solved, there is an urgent need for algorithms with higher robustness for visual keypoint detection and description in extreme scenarios such as high-speed scenes, unmanned aerial vehicle scenes, and foot robots with large motion amplitudes.

Traditional convolutional neural networks possess translational invariance Kauderer-Abrams (2017), a characteristic that proves to be highly valuable in tasks such as image classification and object detection. However, in certain scenarios, such as keypoint detection, pose estimation, and image segmentation, this translational invariance might restrict the model's performance. To address these tasks, researchers have begun to explore the introduction of invariances to transformations like rotation, scale, and viewpoint, aiming to better adapt to the diversity of the real world Zheng et al. (2022). These innovative network architectures and techniques are capable of delivering more accurate and robust results when large rotation transform exists.

The performance improvement brought about by the introduction of positional information is evident in works such as AWDesc Wang et al. (2023), SuperGlue Sarlin et al. (2020), LoFTR Sun et al. (2021). However, positional information contains contextual details that might pertain to various spatially scattered and distinct objects with unknown poses. If we were to combine local rotation-equivariant descriptors with positional information, it wouldn't necessarily achieve complete rotational equivariance in theory. Moreover, in keypoint tasks, thousands of keypoints are often extracted from an image. Therefore, modeling the relationship between the unknown poses of several objects

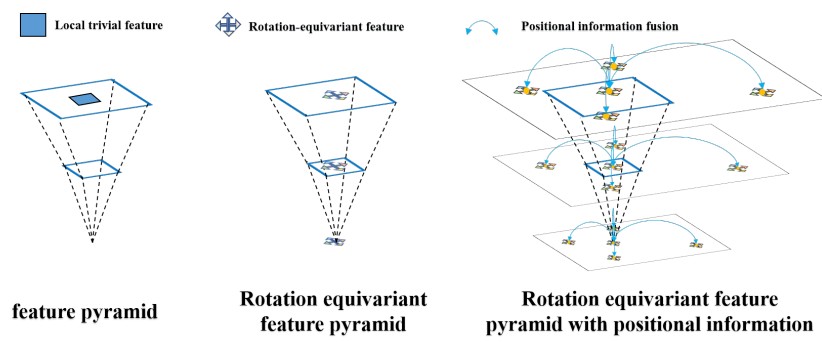

Figure 1: The preview of our method.

in space and the transformations in the image would significantly increase computational complexity and training difficulty for keypoint tasks.

By combining the advantages of rotation-equivariant networks Cohen & Welling (2016) and positional information, we aim to retain the local rotational equivariance of keypoint descriptors while introducing robust positional information. This approach strives to enhance the performance of keypoints in challenging scenarios. It leverages the benefits of both techniques, allowing keypoints to perform as effectively as possible in demanding environments.

In this work, we present an end-to-end framework to simultaneously detect and describe robust keypoints. As shown in Fig. 1, on the left is the commonly used multi-scale feature pyramid for traditional feature extraction. In the middle is the multi-scale feature pyramid with rotation equivariance, while on the right is the multi-scale feature pyramid with rotation equivariance that we have explored and enriched with global positional information. We utilize group-equivariant neural networks for rotation-equivariant local feature extraction. We have efficiently implemented a multi-level fusion of rotation-equivariant features. Taking inspiration from the concept of capturing one-eighth scale local features as introduced in LoFTR Sun et al. (2021), we integrate rotation-equivariant feature maps at that resolution level. Then we carefully design the fusion module of rotation-equivariant feature and positional information. Finally, we proposed the directional uncertainty weighted descriptor loss to train our model. The proposed method is evaluated on several datasets and their rotated version. The experimental results show that our model performs better than many state-of-the-art (SoTA) approaches.

## 2    RELATED WORK

### 2.1    MANUALLY DESIGNED ROTATION-EQUIVARIANT FEATURES

Manually designed features are often crafted to encode feature information based on the pixels surrounding a given pixel. These features are conceived with rotational invariance in mind during their design, taking into account the surrounding pixels, which allows them to be invariant to rotations.

Traditional algorithms Lowe (1999); Mur-Artal et al. (2015); Leutenegger et al. (2011); Alcantarilla et al. (2012); Alcantarilla & Solutions (2011) have distinct advantages in different scenarios, but they all achieve rotation resistance through scale-space analysis, principal orientation assignment, and descriptor computation. This enables them to excel in tasks such as image matching and object recognition under rotation transformations. The choice of the appropriate algorithm depends on specific application requirements, including computational resources, robustness, and real-time performance.

### 2.2    LEARNING-BASED ROTATION-EQUIVARIANT FEATURES

Compared to traditional algorithms, learning-based local feature extraction has demonstrated significant robustness in recent years, particularly in terms of its ability to handle challenges like changes in lighting conditions and varying viewpoints DeTone et al. (2018).

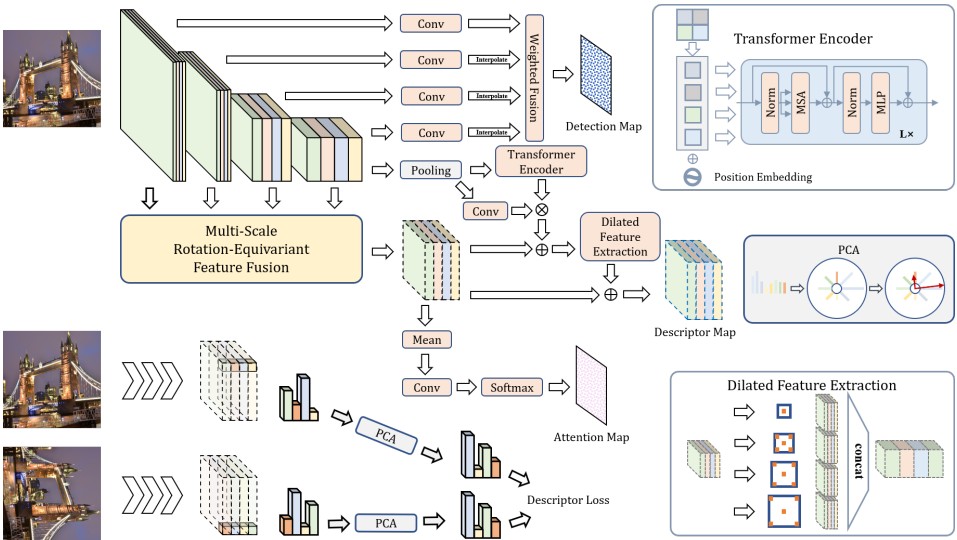

Figure 2: Overview of our pipeline. An image is forwarded to the rotation-equivariant feature extraction backbone. Then these multi-scale feature maps are fed into different weighted fusion module to obtain the keypoint detection map. The deepest feature maps are fed into Transformer Encoder. Then its output and fused multi-scale feature maps are fed into Dilated Feature extraction module to obtain the final descriptor map. At the bottom left of the diagram are a pair of images with rotational relationships. After passing through the aforementioned pipeline, the descriptors obtained from these images undergo PCA (Principal Component Analysis) to compute the principal direction. Subsequently, they undergo circular shifts along the channel dimensions, and the loss function is computed during this process.

Group-equivariant Convolutional Neural Networks Cohen & Welling (2016); Cohen et al. (2018) (G-CNNs) introduce symmetry into convolutional neural networks. The pivotal advantage of G-CNNs lies in their capacity to handle symmetrical data, encompassing operations such as rotations, translations, mirroring, and other symmetries commonly encountered in images. This signifies that the network's architecture and operations take these symmetrical transformations into careful consideration, imparting greater robustness and the ability to capture and harness the inherent symmetry information present in the data.

G-CNNs exhibit remarkable performance in terms of rotation equivariance. Specifically, when the data undergoes a rotational operation,G-CNNs produce output changes that exhibit similar equivariant characteristics. This proves particularly pivotal in numerous computer vision tasks, including but not limited to object detection, image recognition, and image generation. In these tasks, an object's orientation may change, while its essential features should remain invariant.

Additionally, the parameter-sharing capability of group-equivariant networks often translates into reduced data requirements for training. This arises from their ability to glean more generalized features from a smaller set of labeled samples. ReF Peri et al. (2022) utilized the group pooling operation to obtain rotation-invariant feature map from rotation-equivariant feature map. RELF Lee et al. (2023) proposed the group aligning to shift a group-equivariant descriptor refer to its dominant orientation to get a rotation-invariant descriptor.

# 3 METHOD

There are successful cases of positional encoding, which SuperGlue uses for learning based image matching. In Vision Transformer, positional encoding assigns global positional information to each patch of the image, and then calculates the correlation between each patch and all other patches based on self attention. However, local rotation-equivariance and global position information are often incompatible. Because in real-world applications, some objects in an image rotate to varying

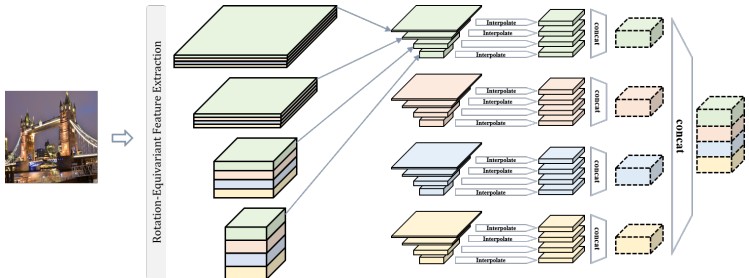

Figure 3: Schematic diagram of the Multi-scale Rotation-Equivariant Feature Fusion module. On the far left is the original multi-level rotation equivariant feature map, and the example in the figure is a schematic feature map of rotation equivariant group convolution with a group size of four. Firstly, feature maps belonging to the same rotation angle on the group are grouped together, and then the multi-level feature maps of each group are interpolated and concatenated together. Finally, the feature maps of each group are concatenated together in the original order, maintaining the rotation invariance of local features during the multi-level feature fusion process.

degrees independently, it is not possible to confuse the rotation of the entire image with the rotation of local features.

## 3.1 MULTI-SCALE ROTATION-EQUIVARIANT FEATURE FUSION

### 3.1.1 EQUIVARIANCE AND INVARIANCE

Given a transformation group $G$, $\mathscr{T}$ is a linear representation of $G$, and $\mathscr{T}'$ is not required to be identical to $\mathscr{T}$. That is to say, $\mathscr{T}$ and $\mathscr{T}'$ may represent the same transform but acts on different space (*e.g.*, spatial coordinate space and group space) Cohen & Welling (2016); Weiler et al. (2018). A group-equivariant function $\mathscr{F}_e : X \longrightarrow Y$ should observe that

$$\mathscr{F}_e(\mathscr{T}(x)) = \mathscr{T}'(\mathscr{F}_e(x)), \tag{1}$$

where $x \in X$, $\mathscr{T} \in G$. While the group-invariant operation $\mathscr{F}_i$ satisfies

$$\mathscr{F}_i(\mathscr{T}(x)) = \mathscr{F}_i(x). \tag{2}$$

### 3.1.2 MULTI-SCALE ROTATION-EQUIVARIANT FEATURE FUSION

In contrast to the single-scale feature extraction approach, FPN Lin et al. (2017) necessitates the construction of the feature pyramid only once, following which these feature maps can be seamlessly transmitted to various task-specific networks. This not only mitigates computational overhead but also augments overall model efficiency.

To make feature extractor to be scaling-invariant, we design the backbone feature extractor with multi-scale feature fusion using FPN. First, the input image is lifted into regular representation space. Then, the rotation-equivariant convolution filters are acting on the regular representation. In this work, four distinct resolution feature maps are combined into a single resolution through inter-polation. Inspired by LoFTR, the concept of capturing one-eighth scale local features is important for local feature extraction. Thus, we integrate rotation-equivariant feature maps at that resolution level.

As shown in Fig. 3, to ensure that rotational equivariance is preserved during the process of feature map fusion, we isolate the rotational group dimensions of the feature maps. Subsequently, we concatenate the feature maps together.

## 3.2 ROTATION-EQUIVARIANT FEATURE AND POSITIONAL INFORMATION FUSION

The multi-head attention mechanism constitutes a pivotal component in deep learning models such as Vision Transformer Dosovitskiy et al. (2020). By amalgamating the outputs of multiple heads, ViT can synthesize information from different locations, enriching the understanding of global relationships. Some learning-based matchers Sarlin et al. (2020); Sun et al. (2021); Wang et al. (2022),

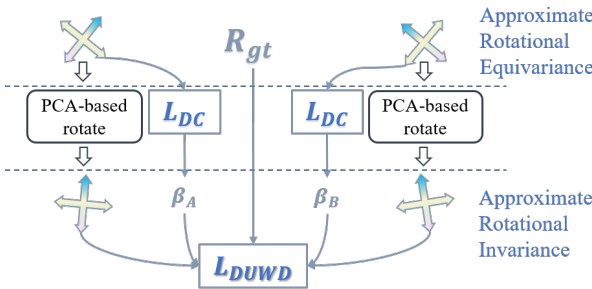

Figure 4: Directional Uncertainty Weighted Descriptor Loss.

benefit the global positional information to enhance the local descriptor in the end2end matching pipeline. Therefore, we also aim to incorporate the powerful global positional information into the framework for local rotation-equivariant keypoint extraction. As shown in Fig. 2, to enhance computational efficiency, the deepest layer of rotation-equivariant features is passed through pooling and then fed into a Transformer Encoder. Additionally, it is weighted by the output of another convolutional pathway. Next, we add the multi-scale fused rotation-equivariant feature maps to it and feed them into a dilated feature extraction module. This is done to compensate for the rotational equivariance loss caused by partially non-rotation-equivariant architectures. Then, the output from the dilated feature module is again added to the multi-scale fused rotation-equivariant feature maps, resulting in the final descriptor map.

### 3.3 DIRECTIONAL UNCERTAINTY WEIGHTED DESCRIPTOR LOSS

In the above description, the feature maps we produce are approximately rotation-equivariant. For keypoints, we expect them to possess (1) robust rotation-equivariant properties, implying that feature descriptors should exhibit high similarity after circular shifts, and (2) good principal direction discriminability, indicating that feature descriptors can perform precise circular shifts to leverage the characteristics of rotation-equivariant features, facilitating keypoint extraction under significant random rotations. We employ Principal Component Analysis (PCA) to estimate the principal direction of rotation-equivariant features.

G-CNNs perform discrete sampling in group space, whereas for rotation-equivariant feature extraction networks, only discrete sampling of 2D rotations is required, with a total of K samples. We believe that the principal direction of a keypoint should be distinct and clear, aligning with the traditional concept of corner points. We select the accumulated maximum value of channels corresponding to angles in the rotation group space as the confidence measure for the principal direction of a keypoint. We expect a higher response value for the principal direction, and thus, we define the confidence of the principal direction for the rotation-equivariant keypoint descriptor as follows:

$$\beta = \frac{\sum_{c=0}^{C-1} D(c, \arg\max_{k} \sum_{c=0}^{C-1} D(k,c))}{\sum_{k=0}^{K-1} \sum_{c=0}^{C-1} D(k,c)}, \tag{3}$$

where $D$ is the rotation-equivariant descriptor, $k \in K$ is the index of discrete rotation group, $c \in C$ is the index of dimensions independent of group channels. Then, similarly, we define a loss function for directional ambiguity as follows:

$$\mathrm{L}_{DC} = -(1-\beta) \sum_{c=0}^{C-1} \log\left(\frac{D(c, \arg\max_{k} D(k,c))}{\sum_{k=0}^{K-1} D(k,c)}\right), \tag{4}$$

and we set the upper limit of the loss function to 20. Due to the discrete nature of the group space, there exists a bias in aligning the principal directions between the feature descriptors to be matched. To minimize the impact of this bias during model training, we take into account a descriptor loss

based on the uncertainty of the principal direction. As shown in Fig. 4, two rotation-equivariant keypoint descriptors for matches from different images, after PCA processing, are transformed into descriptors that are approximately rotation-invariant. Because the rotation group is discrete, even when the ground truth rotations can be obtained during training, there is still an error in the descriptor's orientation after circular shifts along the channels of the discrete rotation group. Therefore, we take into account both the clarity of the descriptor's orientation and the similarity between matched descriptors when designing the following loss function:

$$\mathrm{L}_{DUWD} = \beta_A \beta_B L_{CVtri}(D_A, D_B, w_A, w_B) + L_{DC}(D_A) + L_{DC}(D_B) + CE(Shift(D_A, O_{gt}, D_B)) \quad (5)$$

where $A$ and $B$ are the indexes of the image pairs to be matched, we employed the consistent attention weighted triplet loss $L_{CVtri}$ from AWDesc Wang et al. (2023), $w_A$ and $w_B$ are the attention map of the image pairs to be matched, $CE$ is the cross entropy loss, and $Shift$ can circularly shift descriptors along the dimensions of the rotation-equivariant group based on the ground truth relative rotation $O_{gt}$ between images.

## 4 EXPERIMENT

### 4.1 IMPLEMENTATION

We use E2CNN Weiler & Cesa (2019) to build the rotation-equivariant feature extractor. The group size of SO(2)-equivariant representation we use is 8. Following Wang et al. (2023), the detection ground truth is generated by an off-the-shelf trained SuperPoint DeTone et al. (2018), the loss function we used for detection is weighted binary cross-entropy. The training dataset we used is seleted 118 scenes from MegaDepth Li & Snavely (2018) following D2-Net Dusmanu et al. (2019), and the image are cropped to $400 \times 400$ for training.

We use a computer with an Intel I9-13900K CPU and an NVIDIA GeForce RTX 4090 GPU for training and inference. The learning rate is set to 0.001, the weight decay is set to 0.05, and the batchsize is 12. The training is finished after 28 epochs.

Table 1: Experimental results on the rotated-HPatches dataset.

| Methods | MMA@3 | MMA@6 | MMA@10 |
|---|---|---|---|
| ORB Rublee et al. (2011) | 0.5758 | 0.6554 | 0.6715 |
| BRISK Leutenegger et al. (2011) | 0.7451 | 0.8256 | 0.8412 |
| AKAZE Alcantarilla & Solutions (2011) | 0.7443 | 0.8221 | 0.8413 |
| KAZE Alcantarilla et al. (2012) | 0.7526 | 0.8366 | 0.8578 |
| ReF Peri et al. (2022) | 0.3531 | 0.4115 | 0.4281 |
| RELF Lee et al. (2023) | 0.5110 | 0.5929 | 0.6442 |
| AWDesc Wang et al. (2023) | 0.4554 | 0.5299 | 0.5537 |
| ours | 0.6671 | 0.7973 | 0.8339 |

### 4.2 FEATURE MATCHING EVALUATION

The HPatches Balntas et al. (2017) dataset includes numerous pairs of images that come from different settings, angles, lighting conditions, and types of changes like rotations, translations, scaling, and changes in brightness. This multifaceted composition renders it a challenging dataset, serving as a robust platform for evaluating the performance of algorithms across a spectrum of complex scenarios. It has gained widespread recognition and utilization as a benchmark for assessing and benchmarking image matching algorithms. The rotated-hpatches dataset is generated by randomly rotate the query image of HPatches dataset. Compared to the original HPatches dataset, the rotated-hpatches dataset, which has undergone rotation augmentation, is more challenging. The MMA metric for algorithms with poor rotational robustness experiences a significant decrease in performance on this dataset. We compared ORB, BRISK, AKAZE, KAZE, ReF, RELF, AWDesc, and our method on the rotated-HPatches dataset, as shown in Table. 1. Benefiting from the integration of local rotation-equivariant features, our method exhibits a substantial advantage over other learning-based approaches on the rotated-hpatches dataset. However, we also observe that traditionally handcrafted keypoints with explicitly defined rotational equivariance perform slightly better on

the rotated-hpatches dataset. We believe this is due to the dataset being limited to planar scenes, making it challenging for learning-based algorithms to gain a significant advantage. Therefore, we are considering conducting experiments on real datasets with depth information for camera relative pose estimation.

Table 2: Experimental results on the Megadepth-series datasets.

| | Methods | AUC@5 | AUC@10 | AUC@20 | Prec |
|---|---|---|---|---|---|
| MegaDepth | ORB | 0.26 | 0.96 | 3.06 | 53.75 |
| | BRISK | 8.75 | 16.83 | 27.56 | 73.59 |
| | AKAZE | 8.91 | 17.79 | 29.55 | 77.89 |
| | KAZE | 15.85 | 27.96 | 42.1 | 77.48 |
| | ReF | 27.69 | 46.20 | 64.22 | 86.43 |
| | RELF | 19.03 | 35.06 | 52.45 | 72.10 |
| | AWDesc | 27.69 | 46.20 | 64.22 | 86.43 |
| | ours | 30.42 | 49.46 | 67.23 | 90.09 |
| MegaDepth-Rot90 | ORB | 0.26 | 0.96 | 3.06 | 53.75 |
| | BRISK | 8.75 | 16.83 | 27.56 | 73.59 |
| | AKAZE | 8.91 | 17.79 | 29.55 | 77.89 |
| | KAZE | 17.48 | 29.60 | 43.96 | 77.93 |
| | ReF | 17.61 | 30.98 | 45.98 | 66.51 |
| | RELF | 12.63 | 25.59 | 42.17 | 65.96 |
| | AWDesc | 10.47 | 19.84 | 31.33 | 49.49 |
| | ours | 20.46 | 37.70 | 56.37 | 78.53 |
| MegaDepth-Rot-Rand | ORB | 0.26 | 0.88 | 2.8 | 53.47 |
| | BRISK | 5.11 | 11.1 | 20.42 | 71.52 |
| | AKAZE | 5.54 | 11.81 | 21.53 | 76.24 |
| | KAZE | 15.77 | 28.64 | 43.29 | 78.87 |
| | ReF | 6.9 | 12.32 | 18.37 | 37.73 |
| | RELF | 14.76 | 28 | 43.43 | 65.93 |
| | AWDesc | 10.54 | 19.85 | 28.97 | 49.17 |
| | ours | 16.76 | 31.79 | 49.18 | 74.27 |

### 4.3 POSE ESTIMATION EVALUATION

The MegaDepth Li & Snavely (2018) dataset stands out as an ideal choice for evaluating the performance of image matching and camera relative pose estimation due to several key attributes. It sets itself apart by including depth information, drawing from real-world scenarios, applicability across diverse use cases, and the provision of standardized performance metrics. The dataset not only encompasses depth information but also showcases imagery captured from authentic environments, spanning various applications. Moreover, MegaDepth provides established performance metrics, such as reprojection error and pose error, specifically tailored for assessing the precision of algorithms in estimating camera motion. These metrics serve to quantitatively measure the accuracy of algorithms in the context of camera relative pose estimation.

The estimated pose is determined from the matches by calculating the essential matrix. Subsequently, we follow the same methodology as described in the reference to compute the Area Under the Curve (AUC) of pose error at specific thresholds (5 degrees, 10 degrees, 20 degrees). In this context, pose error is defined as the greater of the angular errors in rotation and translation. Since the AUC of pose error incorporates RANSAC for pose estimation, it can yield accurate pose estimates by filtering out numerous mismatches. However, it's worth noting that the AUC of pose error may not provide a comprehensive assessment of the matching method due to its reliance on RANSAC. To overcome this limitation, match precision takes into account all matches, including mismatches, offering a more holistic evaluation of the method's performance. Therefore, we also employ the approach outlined in SuperGlue to calculate matching precision and utilize it as an additional metric for assessing the quality of correspondence matching.

We further augmented the MegaDepth dataset in two different ways: by applying random rotations in multiples of 90 degrees and by applying random rotations at arbitrary angles. These augmented datasets are referred to as MegaDepth-Rot90 and MegaDepth-Rot-Rand, respectively. We conducted experiments on MegaDepth, MegaDepth-Rot90, and MegaDepth-Rot-Rand, as indicated in the table. Our approach exhibited the best performance on the MegaDepth, MegaDepth-Rot90, and MegaDepth-Rot-Rand datasets. The experimental results are shown in Table. 4. It is evident that our approach demonstrates superior performance in all three modes of the dataset and significantly outperforms traditional algorithms. Furthermore, due to the introduction of global positional infor-

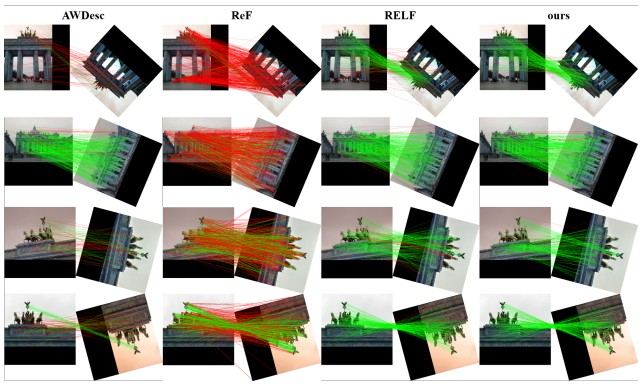

Figure 5: Visualization of matches on the MegaDepth-Rot-Rand dataset.

mation in our method, perfect rotational equivariance cannot be achieved. Therefore, as the rotation difficulty of the dataset increases, the algorithm's performance decreases. Traditional algorithms like KAZE, designed to be sufficiently robust, although not outperforming our method overall, exhibit resilience to rotations. We present visualizations of the matching results on MegaDepth for AWDesc, ReF, RELF, and our method in Fig. 5, where green lines represent correct matches, and red lines represent incorrect matches. From the visual results, it is evident that our method exhibits richer matches and higher accuracy.

Table 3: Experimental results on the YFCC100M-series datasets.

|  | Methods | AUC@5 | AUC@10 | AUC@20 | Prec |
|---|---|---|---|---|---|
| YFCC100M | ORB | 0.05 | 0.23 | 0.87 | 24.22 |
|  | BRISK | 3.38 | 7.12 | 12.69 | 53.35 |
|  | AKAZE | 1.44 | 3.4 | 6.86 | 51.04 |
|  | KAZE | 4.55 | 9.61 | 17.64 | 59.45 |
|  | AWDesc | 26.07 | 44.87 | 63.15 | 72.66 |
|  | ReF | 8.75 | 18.25 | 31.44 | 24.88 |
|  | RELF | 9.19 | 18.93 | 31.9 | 35.81 |
|  | ours | 26.25 | 44.9 | 62.87 | 74.44 |
| YFCC100M-Rot90 | ORB | 0.07 | 0.19 | 0.77 | 24.66 |
|  | BRISK | 2.97 | 6.62 | 12.3 | 53.2 |
|  | AKAZE | 1.44 | 3.4 | 6.96 | 50.6 |
|  | KAZE | 4.61 | 9.36 | 17 | 59.19 |
|  | AWDesc | 8.49 | 16.02 | 25.2 | 34.58 |
|  | ReF | 8.6 | 18.09 | 31.04 | 24.88 |
|  | RELF | 7.71 | 16.1 | 28.13 | 32.72 |
|  | ours | 11.95 | 23.89 | 38.18 | 50.58 |
| YFCC100M-Rot-Rand | ORB | 0.07 | 0.22 | 0.92 | 25.05 |
|  | BRISK | 3.56 | 7.15 | 12.7 | 53.7 |
|  | AKAZE | 1.35 | 3.15 | 6.58 | 50.87 |
|  | KAZE | 4.2 | 9.47 | 17.59 | 59.34 |
|  | AWDesc | 8.45 | 15.95 | 24.86 | 33.4 |
|  | ReF | 2.24 | 5.32 | 10.57 | 12.8 |
|  | RELF | 4.46 | 11.25 | 21.59 | 29.52 |
|  | ours | 11.86 | 24.1 | 39.58 | 52.29 |

The evaluation approach applied to the YFCC100M Thomee et al. (2016) dataset closely mirrors that of MegaDepth. In line with SuperGlue's methodology Sarlin et al. (2020), we carefully choose identical pairs from the YFCC100M dataset for testing, ensuring an equitable basis for comparison. Subsequently, we employ the same evaluation methodology to calculate the Area Under the Curve (AUC) of pose error at specified thresholds (5 degrees, 10 degrees, 20 degrees). Furthermore, we can derive the precision of the matches as an additional evaluation metric. Our approach exhibited the best performance on the YFCC100M, YFCC100M-Rot90, and YFCC100M-Rot-Rand datasets. Across the three modes of the YFCC100M dataset, learning-based methods further widen the gap between them and traditional algorithms. Our approach significantly outperforms other algorithms.

Table 4: Results of ablation study on the Megadepth-series datasets.

|  | Methods | AUC@5 | AUC@10 | AUC@20 | Prec |
|---|---|---|---|---|---|
| MegaDepth | ours | 30.42 | 49.46 | 67.23 | 90.09 |
|  | ablation1 | 25.15 | 42.14 | 58.87 | 83.68 |
|  | ablation2 | 27.64 | 45.28 | 61.42 | 72.91 |
| MegaDepth-Rot90 | ours | 20.46 | 37.70 | 56.37 | 78.53 |
|  | ablation1 | 13.18 | 24.55 | 39.03 | 43.32 |
|  | ablation2 | 10.56 | 19.60 | 31.69 | 38.45 |
| MegaDepth-Rot-Rand | ours | 16.76 | 31.79 | 49.18 | 74.27 |
|  | ablation1 | 9.97 | 19.85 | 33.27 | 39.38 |
|  | ablation2 | 9.29 | 18.91 | 31.66 | 37.11 |

## 4.4 ABLATION STUDY

We explored the effectiveness of the proposed method in the ablation experiment. Our feature fusion process is:

$$DFE(F + (TE(C_4) * Conv(C_4))) + F, \tag{6}$$

ablation1 represents the following feature fusion process:

$$DFE(F + (TE(C4) * Conv(C4))) + F + (TE(C_4) * Conv(C_4)), \tag{7}$$

where $DFE$ is the dilated feature extraction, $F$ is the fused feature map of multi-scale rotation-equivariant feature fusion, $TE$ is the Transformer encoder, $C_4$ is the last feature map of feature backbone, $Conv$ is the convolution operation, $*$ is the element-wise multiply operation in channel dimension. Ablation2 represents the following feature fusion process:

$$F + DFE(TE(C_4) * Conv(C_4)). \tag{8}$$

Ablation1, to some extent, increases the proportion of global position information, which is not conducive to expressing local rotation and other variable information. Ablation2 does not incorporate the fused multi-level rotation-equivariant features into the dilated feature extraction. Instead, it fuses the global position information processed by multi-scale hole convolution with the multi-level rotation-equivariant features, resulting in a lack of a good connection between the multi-level rotation-equivariant features and the global position information.

From the experimental results, it can be seen that ablation1 performs weaker than other fusion methods on the dataset without rotation augmentation. However, due to its multi-level rotation-equivariant features and the higher number of connections between global position information fusion, it performs better than ablation2 on the dataset with 90-degree augmentation.

## 4.5 RUNTIME ANALYSIS

We compared the running speed of our method with other methods on the rotated patches dataset, which involved keypoint extraction and matching of a pair of images. The AWDesc runtime is 0.3106 seconds, while ours is 0.4785 seconds. The results show that our method significantly enhances anti-rotation performance while maintaining acceptable computational complexity.

## 5 CONCLUSIONS

In this article, we have addressed and researched the problem of image feature extraction, leading to the design of a more robust feature extractor architecture that surpasses existing methods on multiple datasets. Specifically, we utilized a rotation-equivariant feature pyramid to provide locally rotation-equivariant feature information, along with a multi-head attention mechanism for adaptively fusing positional information in the features. Furthermore, we introduced the directional uncertainty weighted descriptor loss to enhance the model's robustness. Looking ahead, we plan to explore techniques such as model quantization and knowledge distillation to accelerate image feature extraction while maintaining model performance.

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
