# OpenReview forum: "Rotation-Equivariance and Position Encodings for Enhancing Local Descriptors"
_ICLR.cc/2024/Conference — Submitted to ICLR 2024_

### Official Review · Reviewer_rhbE · 2023-10-30

**Soundness:** 3 good
**Presentation:** 2 fair
**Contribution:** 3 good
**Rating:** 5
**Confidence:** 5

**Summary:**

The paper introduces a novel approach to extend the rotation equivariance property in local features (keypoints, desctiptors). Keypoint detection and description are essential in various downstream tasks of computer vision. The authors proposes the multi-scale rotation-equivariant feature fusion and the directional uncertainty weighted descriptor loss. The feature fusion stage isolates and concatenates the group dimension to ensure rotation equivariance. The paper also conducts a comparative analysis of classical equivariant descriptors and group-equivariant network-based descriptors on a standard image matching benchmark.

**Strengths:**

Motivation of the Problem: The paper addresses a crucial problem in computer vision, which is the extraction of rotation-equivariant local features such as keypoints and descriptors. This problem has broader applications beyond just image matching, making the research highly relevant in the computer vision.

Soundness of Methodology: The utilization of multi-scale features to enhance scale robustness is well-founded. Additionally, incorporating a desctiptor loss by shifting in the direction using principal components is sound.

**Weaknesses:**

Improper citation. When citing a paper in ICLR format, citing in the middle of a sentence makes the sentence incomplete. This should be corrected to become a natural sentence.
It is not convinced to mention illumination invariance in the first paragraph of Section 1. This paper does not directly address illumination. (e.g., Tang et al., 2019)
In the first paragraph of Section 1, you give examples of image matching, 3D reconstruction, and object tracking, but the actual paper you cite is a Roxf/par paper on image retrieval. Additional citations of related papers are needed.

In section 2 related work, authors do not write a list of existing papers. Discuss the similarities and differences between them and this paper.
In addition, authors should discuss about a paper for Learning-based rotation-equivaraint keypoint detectior [1].

[1] Self-Supervised Equivariant Learning for Oriented Keypoint Detection (Lee et al., CVPR 2022)

In section 3, authors replace the subheadings of sections 3-1. For example, feature extraction should be discussed first, followed by feature fusion, for a more natural flow.

In section 3-3, authors should add citations to two papers that perform a principle component shift similar to the proposed directional uncertainty weighted descriptor loss function. [2], [3]

[2] Self-supervised Learning of Image Scale and Orientation (Lee et al., BMVC 2021)
[3] Learning Soft Estimator of Keypoint Scale and Orientation with Probabilistic Covariant Loss (Yan et al., CVPR 2022)

Insufficient explanation of F/T captions: please add sufficient explanations to the captions of Figures 3,4,5, and Tables 1,2,3.

In section 4, authors should add a separate section for the description of the dataset. For example, in MegaDepth-series, YFCC100M, please explain what the series is separately.

Needs an ablation study: authors should add information about the performance gain of each component for the multi-scale feature fusion method proposed in the paper, the number of pyramids, and the loss function.

**Questions:**

Please see the weaknesses section to address my questions.

---

> ### Author Response · Authors · 2023-11-22
> **rebuttal**
>
> Thank you for your valuable feedback.
> We addressed issues related to image retrieval literature.
> We have supplemented ablation experiments for the proposed feature fusion method, demonstrating that to achieve the best results, it is crucial to retain local rotational equivariance of feature maps while obtaining global position information. A smooth integration process in feature fusion is essential for optimal performance.
> We are still actively conducting additional experiments and making further revisions to the paper. Thank you very much for your valuable feedback.

---

### Official Review · Reviewer_FmdE · 2023-10-30

**Soundness:** 3 good
**Presentation:** 3 good
**Contribution:** 3 good
**Rating:** 6
**Confidence:** 3

**Summary:**

This paper proposes a new learning based local descriptors, which restorts to position ecodings and fuses multi-level rotation equivariant features.
The proposed method can maintain the overall performance of descriptors, while obtaining rotation-equivariance to deal with the large reotation.
The experiments show the proposed method improves the performance of keypoint extraction and description.

**Strengths:**

1. The paper is generally well-written. The figures illustrate the framework of the proposed method clearly.
2. The idea to fuse multi-scale feature is reasonable and the multi-head attention is also a reasonable tool to fuse features.
3. The authors perform experiments to compare the proposed method with most of the mainstream descriptors and achieve the state-of-the-art.

**Weaknesses:**

1. The whole framework is a litte complicated. The authors do not explain the necessity of each module clearly, neither perform abation study to show each module's effect.
2. The efficiency of descriptior extraction is also important for some application, but the authors do not show the running time of the proposed method in the experiments.
3.In the description of figure2, all "are fed" is written as "are feed".

**Questions:**

Could the authors compare the running time of the proposed method and other descriptors?

---

> ### Author Response · Authors · 2023-11-22
> **rebuttal**
>
> Thank you for your valuable feedback. We have conducted additional ablation experiments and comparative experiments on runtime.
>
> We performed ablation experiments on the proposed feature fusion method, indicating that for optimal results, it is crucial to retain the local rotational equivariance of feature maps while obtaining global position information and ensuring a smooth integration process in feature fusion.
>
> We calculated the average runtime for feature point extraction and matching of a pair of images on the rotated-hpatches dataset: AWDesc: 0.3106 seconds, ours: 0.4785 seconds. The experiments demonstrate that our method exhibits high real-time performance.
>
> We have corrected a spelling error in Figure 2.

---

### Official Review · Reviewer_QDcf · 2023-10-31

**Soundness:** 2 fair
**Presentation:** 2 fair
**Contribution:** 2 fair
**Rating:** 3
**Confidence:** 5

**Summary:**

This paper studies rotation equivariance for descriptor learning in image matching scenario. They proposed new model and training for learning robust descriptor following the previous work E2CNN with equivariant steerable cnns.

Rotation-equivariant feature pyramid (FPN) is used and local rotation-equivariant feature information is captured with a multi-head attention mechanism fusing positional information.

Multiple datasets are compared to show the improvement of the proposed method.

**Strengths:**

The proposed direction is important in descriptor learning research. Rotation equivariance is not the main stream of the image matching at the moment, but the authors proposed a way through PCA feature direction for learning such losing property via a rotated pair images. I find the idea quite novel.

The results in Figure 5 shows improvement over some descriptors over severely rotated image pairs. Multiple datasets are studied and compared.

Most real-world dataset shows improvement over the baselines.

**Weaknesses:**

Despite I understand the idea behind the rotation equivariance descriptor design, the paper hardly justifies how the performance improves the state-of-the-art.

For example, the authors claim "If we were to combine local rotation equivariant descriptors with positional information, it wouldn’t necessarily achieve complete rotational equivariance in theory" while referring AWDesc Wang et al. (2023), SuperGlue Sarlin et al. (2020), LoFTR Sun et al. (2021) In the experimental section, only AWDesc is truly compared while SuperGlue and LoFTR and the widely accepted baselines nowadays.

There is no comparison regarding the proposed descriptor improves any classical matcher such as RANSAC variants (e.g. MAGSAC, MAGSAC++, AdaLAM, PROSAC...) or learnable matcher such as SuperGlue, ACNe, or NG-RANSAC.

There is also no clear discussion nor proof how this improves against dense matching method such as LoFTR.

Many SOTA descriptors are not being directly compared such as Superpoint, D2-Net, R2D2, ...

I am also very surprised to see the proposed method is losing to 2011 methods such as BRISK in Table.1.

Some results are positive, but they are not enough to complete the claim of how this rotation equivariant complement the weakness of 1) SOTA descriptors, and 2) modern matcher research which is the main research domain of the community.

Minor:

Some pictures and descriptions are not clear enough. (e.g. Fig.1)

**Questions:**

I do like the direction and effort, if the author can clarify how their work achieve or improve the above weakness I mentioned I would be happy to see the rating go higher.

1. Why the proposed method is weaker than most of 2011-2012 methods in Table.1 if you really solve the rotation equivariance problem? I found the discussion of planar dataset is not acceptable.
2. How exactly does this descriptor learning being used with modern matchers (classical, deep learning)?
3. How does this method relates to the popular dense matching and why is it better in a certain way?
4. Show me some actual results to prove the above points rather than just arguments would be the strongest case, but that also requires the authors basically rewriting the whole experimental section. Therefore, I am leaning towards to reject this paper but encouraging the authors to complete the full thing before re-submission.

**Details Of Ethics Concerns:**

No ethics concerns

---

> ### Author Response · Authors · 2023-11-22
> **rebuttal**
>
> Thank you for your valuable feedback. We have conducted additional ablation experiments on the proposed feature fusion method, indicating that for optimal results, it is crucial to retain the local rotational equivariance of feature maps while obtaining global position information and ensuring a smooth integration process in feature fusion.
>
> Our method falls into the category of feature point descriptors, similar to SuperPoint and AWDesc, while SuperGlue and LoFTR are considered matchers. When evaluating descriptor performance, we often use the default RANSAC method in OpenCV for camera relative pose estimation on datasets. Therefore, methods like MAGSAC, MAGSAC++, AdaLAM, and PROSAC are not within the scope of feature point descriptor research.
>
> Methods like LoFTR, which focus on dense matching, follow a detector-free end-to-end matching architecture where feature point descriptors are not explicitly extracted, making them less flexible for downstream applications. In our track, which emphasizes feature point descriptors, the extracted feature points are explicit and flexible.
>
> We appreciate your suggestion to conduct experiments using the combination of 'ours' and 'SuperGlue,' but in the track of feature point descriptors, this approach may not be the most direct and effective experiment.
>
> Regarding the rotation-equivariant capability of methods like Superpoint, D2-Net, and R2D2, which is mentioned in DISK and RELF papers, we compare our method with AWDesc under normal conditions without significant rotation and with RELF under substantial rotation. We can supplement experiments for Superpoint, D2-Net, and R2D2 in the appendix.
>
> Traditional methods like BRISK strike a balance between speed and performance, while KAZE is a robust method for rotation-equivariant, similar to SIFT. We chose KAZE for experimental convenience (due to SIFT copyright).

---

### Official Review · Reviewer_JePm · 2023-11-01

**Soundness:** 3 good
**Presentation:** 2 fair
**Contribution:** 2 fair
**Rating:** 6
**Confidence:** 4

**Summary:**

This paper describes an improvement on typical ML descriptors by 1) replacing canonical convolution with group equivariant version in feature extraction, and 2) feeding extra positional information to local descriptor extraction. Performance improvement are presented on three common local detection/descriptor benchmarks.

**Strengths:**

This paper combines two already explored ideas, namely rotation equivariant local descriptor and global position embedding, and fused them together into a seemingly more powerful detection and description solution. Results from 3 benchmarks are aligned and suggest that the final solution indeed captures beneficial information than baselines, that results to better matching and estimation in the end.

**Weaknesses:**

The two major comments are:
- lack of ablation and analysis. The paper introduces two major improvements to existing local descriptor extraction algorithm, however it's surprising to not seeing a ablation on the delta introduced by each change. Especially since there are already baseline methods using group equivariance convolution as well, it could easily lead readers to think the extra improvement is coming from global positional embedding, which arguably could be injected not as part of local descriptor, but during the matching process. There's no doubt that doing a through ablation requires more work, but for this type of paper where multiple features are introduced, it will be extremely valuable to cover that.
- presentation. Try to focus more on the main contribution of the paper (with more text / figures). The current version contains a lot of introduction level content, e.g., on what is feature pyramid network / what is multi-head attention, which is OK when publishing to general audience that are not working on Computer Vision field, but could be redundant (and diminishing YOUR contribution) as a ICLR submission. Besides, there are some typo and grammar fixes worthy of doing.

**Questions:**

As mentioned above, an ablation on the two major features introduced in this paper on some example dataset, and keep the content more focused (so you have some room for ablation as well).

---

> ### Author Response · Authors · 2023-11-22
> **rebuttal**
>
> Your valuable feedback has been immensely helpful to us. We have conducted additional ablation experiments on the proposed feature fusion method, demonstrating that preserving the local rotational equivariance of feature maps while obtaining global position information and ensuring a smooth integration process in feature fusion is essential for achieving optimal results.
>
> We have carefully revised the presentation of the article, removing some fundamental concepts that occupied significant space.

---

### Meta-Review · Area_Chair_To11 · 2023-12-05

**Metareview:**

This paper proposes a fusion of local rotation-equivariant descriptions with globally encoded positional information and a directional uncertainty weighted descriptor loss. Experiments were performed on three datasets.

The paper received one “reject” rating, one “marginally below the acceptance threshold” rating, and two “marginally above the acceptance threshold” ratings.

The main problems given by the reviewers are as follows:
1)	The presentation needs great improvement.
2)	More ablation and analysis are needed.
3)	The paper hardly justifies how the performance improves the state-of-the-art.
4)	Some pictures and descriptions are not clear enough.
The authors gave responses but the revisions cannot be found.

Besides, I would like to say the authors don’t know the local rotation-equivariant descriptors well. Some main rotation-equivariant descriptors are not cited, such as:
i)  GIFT: Learning Transformation-Invariant Dense Visual Descriptors via Group CNNs.
ii)  Beyond Cartesian Representations for Local Descriptors.

Moreover, the popular rotation equivariant descriptor SIFT wasn’t compared in the experiments. The authors said this is due to the SIFT copyright. I think this is not a problem for scientific researches not to mention that the commercial copyright has been free.

Based on the above comments, the decision was to reject the paper.

**Justification For Why Not Higher Score:**

The main problems given by the reviewers are as follows:
1)	The presentation needs great improvement.
2)	More ablation and analysis are needed.
3)	The paper hardly justifies how the performance improves the state-of-the-art.
4)	Some pictures and descriptions are not clear enough.
The authors gave responses but the revisions cannot be found.

Besides, I would like to say the authors don’t know the local rotation-equivariant descriptors well. Some main rotation-equivariant descriptors are not cited, such as:
i)  GIFT: Learning Transformation-Invariant Dense Visual Descriptors via Group CNNs.
ii)  Beyond Cartesian Representations for Local Descriptors.

Moreover, the popular rotation equivariant descriptor SIFT wasn’t compared in the experiments. The authors said this is due to the SIFT copyright. I think this is not a problem for scientific researches not to mention that the commercial copyright has been free.

**Justification For Why Not Lower Score:**

N/A

---

### Decision · Program_Chairs · 2024-01-16

Reject